# Development and validation of a Community Resilience Scale for Youth (CRS-Y)

**Sofia Marques da Silva** *, **Gil Nata, Ana Milheiro Silva, Sara Faria**

Centre for Research and Intervention in Education, Faculty of Psychology and Educational Sciences, Department of Sciences of Education, University of Porto, Porto, Portugal

* sofiamsilva@fpce.up.pt

**Data Availability Statement:** Data files are available from the Zenodo database. (https://zenodo.org/record/6557048#.YoPVfejMKUI, DOI: 10.5281/zenodo.6557048).

## Abstract

The purpose of this article is to present the development and validation of a Community Resilience Scale for Youth (CRS-Y) among a Portuguese sample of nearly 4000 young people growing up in regions on the border with Spain. The scale was developed for young people to assess their perception of the resilience of regional communities in terms of positive development and purposeful experiences for young people. Resilient communities, under a social ecological approach, are those able to move forward on social change and transformation. This concept is especially remarkable in more challenging contexts such as border regions of mainland Portugal which are characterised by economic, social, educational, and cultural disadvantages while discovering possibilities of resilience through promising local dynamics. A multi-step approach was used to develop this scale of 12-item scale. Items were generated based on an in-depth literature review and research previously conducted with young people in these contexts. The overall sample was randomly divided into two subsamples of 1828 and 1735 young people each. Principal component analysis was performed with one of the subsamples and yielded a three-factor structure, explaining 61.5% of the total variance. Confirmatory factor analysis performed on the second showed good fit indexes. Furthermore, internal consistency of the three proposed components, gauged either by Cronbach's alpha or McDonald's omega, indicated good reliability. Given the results, the CRS-Y is a valid and reliable tool showing adequate psychometric properties. This scale will be useful for schools and policy makers at the local level. Indicators such as the promotion of opportunities to participate and be recognised, collective trust and the promotion of shared values and protection are relevant in assessing regional communities' resilience and informing youth policies.

## 1. Introduction

Most border regions in the mainland Portugal offer less than ideal circumstances for young people (14–24). Regions on the border with Spain, particularly those located in inland and mainly rural areas are less developed regions, with few opportunities in education, participation and employment, and suffer from population decline [1, 2]. As stated in the Eurostat

**Funding:** European Regional Development Fund (ERDF), through the Norte Portugal Regional Operational Programme (NORTE 2020), under the PORTUGAL 2020 Partnership Agreement, and the Portuguese Foundation for Science and Technology, IP (FCT) [grant no. POCI-01-0145-FEDER-029943 / PTDC/CED-EDG/29943/2017] (SMS); the Portuguese Government, through the FCT, under CIIE's multi-annual funding [grants no. UID/CED/00167/2019, UIDB/00167/2020 and UIDP/00167/2020] (CIIE; GN is supported by the FCT, under the Scientific Employment Stimulus Individual Programme [grant no. CEECIND/00646/2018]; AMS was supported by the FCT and by the European Social Fund, though the Human Capital Operational Programme (POCH) from PORTUGAL 2020 [grant no. PD/BD/128118/2016].

**Competing interests:** The authors have declared that no competing interests exist.

Regional Yearbook, this situation affects the pathways of young people that choose "to leave the region in which they grew up so they could continue their studies or look for alternative and perhaps more varied work. It was particularly apparent across sparsely-populated regions in Greece, Spain and Portugal" [2] (p. 26). Attracting and encouraging young people to stay or to return have been challenges at different levels of governance in which it has been sought to create better conditions for new generations. Statistics on purchasing power at the municipality level, developed by the Portuguese National Institute of Statistics, indicated that in 2017 inland border regions were among those with the lowest purchasing power per capita or purchasing power percentage (PPP) [3]. Only two, from thirty-eight borderlands municipalities were above 25% percentage of purchasing power, an indicator derived from the indicator of purchasing power per capita and reflects the importance of purchasing power manifested daily in each municipality or region in the total of the country for which the PPP is 100% [3]. However, these regions seem to develop situated approaches that may indicate resilience characteristics. Previous research [4–6] has highlighted local strategies to explore and recognise strengths and to promote opportunities and support, especially for young people.

A community resilience approach is a multi-systemic and dynamic social process which is partly determined by the capacity for collective action [7]. It involves the possibility of social change and transformation [8], and our motivation is to understand how communities in deprived (border) regions activate change and promote positive development and solid pathways among young people. Positive youth development focuses on strategies to improve and promote the full potential of young people so that they can "learn and thrive in the diverse settings where they live" [9] (p. 13). The positive development of young people can be enhanced through trust and social support [10], by creating opportunities to develop skills for participation, include diversity and to shape purposeful futures.

The distinction between dimensions that contribute to communities' resilience and dimensions of resilience is an ongoing debate [11, 12]. With this scale we aim to contribute to this discussion, namely by exploring young people's perceptions about community resilience: how they evaluate the availability of opportunities for their development and empowerment in their regional communities and how they assess intentional collective action and infrastructure, which are indicators of community competence [13]. That is, we aim to measure the young people's perception of their communities' resilience regarding their lives as youngsters.

Resilience can be gauged in different ways and depends primarily on theoretical imports. We are aligned with a multisystemic and social-ecological understanding of resilience that develops within wider social ecologies to which social actors belong [14, 15]. Consequently, resilience is conceptualised here as a collective and community characteristic, rather than an individual response to instability. Therefore, one crucial way to assess a community's resilience is through the perspectives of its members. Several studies on resilience involve local and cultural situated knowledge and *emic* perspectives of social actors to develop or refine attributes of the construct [16]. Therefore, a variety of interlocutors were considered when developing measures regarding resilience, and particularly, community resilience, such as coastal or rural communities [16, 11], community stakeholders and the indigenous population in Australia [17], Afghan young people [18–20], and children and young people in Ungar and Liebenberg [21]. These cases underscore the relevance of building measures grounded on place based and *emic* understandings.

The aim of this scale, which was developed using a literature review and prior research with young people growing up in border regions of Portugal, is to assess the existence of attributes of resilience through perceptions of young people in each respective community.

Resilience has been associated with disaster response and recovery and is often subject to criticisms due to normative implications and of the danger of neoliberal forms of

governmentality [22], reproducing the causes of the instability and not challenging the main roots of the problems [23]. However, the concept of resilience is used at large and the scrutiny of how it is used is relevant. The concept is still needed to understand attributes of resilience in contexts that would be considered to be lacking resilience and to design a future which considers insiders insights into solutions.

The communities where we want to apply the scale did not suffer from a particular disaster from which they need to recover, but most have suffered from long-term structural inequalities and social and spatial injustice regarding educational and social opportunities, trajectories, and prospects for young people. In this sense, we underscore the pertinence of moving the focus of attention to processes and to social ecologies to better understand how to work with different communities that have been dealing with such conditions, leading, for example, to the imperative of leaving, a process of youth socialisation to leaving their regions [24, 25]. In keeping with Clark and Ungar [15], we think we may contribute to contextualising the meaning of resilience by remembering social ecologies and individuals' interaction to promote change at multiple systemic levels and not only to produce resilient citizens [15, 26] or, as in early studies of resilience, focus on the exceptional [27]. Resilience entails the engagement of different systems. In our case this means that if a community has specific attributes of resilience, such as empowering cultural or social practices, the provision of resources towards youth development and trajectories may impact the regional development, fixation of population, etc. As resilient communities also result from a collective commitment, they may encourage more inclusive approaches and maximise sustainability and life conditions. This standpoint may be helpful in informing policies and practices in order to work towards communities that make conditions for personal and collective development available. The CRS-Y is a youth-centred scale, focusing on communities' resilience towards youth development, and designed to highlight young people's perceptions on their communities. The knowledge resulting from sing the scale may have different purposes, as it may be used by communities for self-assessment and to ground policies and practices.

## 1.1. Resilience

Resilience has been considered a boundary object or a bridging concept [28–30], with an explanatory power shared among research communities, disciplines and fields that furthers collaboration and communication [28].

In the social sciences, investments in research into human development have focused on assessing resilience as an individual characteristic with a clear dominance of the individual-centred model [14]. This is consistent with the psychological tradition [12], considering resilience as a competence of individuals to respond to events.

While acknowledging debates about the limits and conservative nature of the concept of resilience [31], we think it is a substantive construct to study responses in different social and ecological contexts [7, 32]. Cutter [33], in one of the works selected by Ungar [34] as part of the target sample of their synthesis and meta-synthesis of resilience literature, identified the following variables for the study of resilience: "educational equality, income, social capital, health access, mitigation plans, religious affiliations, community aspirations, emergency management assets, mitigation activities, infrastructure and buildings" [34] (p. 5).

Research has demonstrated that when understood as a systemic interdependent interaction and integrating social determinants [14, 34, 35], resilience is highly relevant for studying communities in adverse or extreme situations. In developing an ecological understanding of resilience, Ungar [14, 35] assumes that in studying resilience, we must first consider the context and then the individual, proposing resilience as a multisystemic concept [27]. Norris and

colleagues' [13] proposal, while focusing on a community's propensity to respond to adverse events, extended the discussion of the capabilities associated with individuals to communities and societies. Nevertheless, research indicates that a resilient community will promote individual resilience [36], individual effectiveness and collective efficacy [37].

## 1.2. Resilient communities

The notion of resilient communities in terms of crisis resolution, especially economic crises, was developed by Campanella [38] and Callaghan and Colton [39] while working in the field of environmental studies. Although Callaghan and Colton defined resilient communities as "those that are able to absorb and/or adapt quickly to change and crisis" [39] (p. 932), they did not understand resilience as the ability to respond only to immediate and short-term needs. This last aspect is fundamental to this study since many social challenges, such as structural inequalities associated with youth development, are in most cases critical over a long period and require robust systemic approaches.

The complexity of how communities function and on what scale makes it difficult to grasp the overall purpose and goals of a particular community [39] and to understand what might affect the development of a community's resilience. Familiarity with long-term and structural constraints that are difficult to analyse and often perceived as insoluble (e.g., social disadvantage; ethnic discrimination) requires a different understanding of community practices, policies and governance. An indicator of a resilient community is its capacity to define a set of goals and commitment to develop persistent actions that respond to disadvantaged situations. Therefore, it is relevant to consider the different characteristics of a resilient community when reacting to an atypical stressor, but also its competence to evolve when faced with a structural and sometimes continuous source of vulnerability.

The concept of community capital is consistent with the ecological approach to resilience [7] through the integration of different types of capital (social, public structural, cultural, etc.). While there are limitations to the ecological approach, it is still effective when it comes to social relations [31] and for understanding different ecosystems that may compose a community.

Social capital, a resource of resilience, is a type of capital developed around shared values and trust, reciprocity and collective action [7, 40–43]. As defined by Adger [7],

> social capital describes relations of trust, reciprocity, and exchange; the evolution of common rules; and the role of networks. It gives a role to civil society and collective action for both instrumental and democratic reasons and seeks to explain differential spatial patterns of societal interaction. (p. 389)

Social capital appears to be fundamental in building collective efficacy and resilient communities, especially in bridging social capital associated with the idea of process, consolidation of social networks, and provision of support, comfort and trust [44]. Norris and colleagues also analysed the value of networking associated with resilience [13, 45].

In addition to social capital, other forms of capital, such as economic, political, natural [41], cultural and educational, are noteworthy in understanding how communities facing change and challenges develop and organise themselves. Our approach to the concept of resilient communities is to recognise that communities have weaknesses and strengths:

> Resilience of the community itself involves the dynamics of the social response to challenges that threaten to damage or destroy the community. These dynamics may involve adaptations and adjustments of individuals, groups and organisations within the community

(seen as components of the community as a system), as well as interactions of the whole community with its surrounding environment, especially including other social, economic and political entities. [46] (p. 66)

## 1.3. Resilient communities to support young people

This study aims to develop and validate a youth-centred community resilience scale in border regions. We developed a scale to assess the resilience of communities in supporting young people's development and life prospects in border regions that are in general rural, peripheric and economically vulnerable. As Panter-Brick [18] and Panter-Brick and colleagues [19, 20] have argued, the study of resilience benefits from a deeper knowledge of the context and culturally based measures. This aspect was also a justification for Ungar and Liebenberg's [21] development of a culturally sensitive measure of the resilience of young people, the Child and Youth Resilience Measure (CYRM).

Our aim was to investigate how young people, as cultural insiders [16] perceive and assess attributes of resilience as social responses of their regional communities to the needs and problems that might threaten their present and future opportunities.

Following the literature, we have considered the inclusion of items that would help us to understand the level of trust that young people have in their community. Trust has been considered an important indicator in promoting community engagement. Di Napoli et al. [47] believe in "community trust as a composite indicator used to measure community opportunities, as perceived by citizens" (p. 551). The way a young person responds to some items shows the degree of trust in the community, either in items that are more related to action or future orientations [47].

Social actors in Portuguese border regions, as they face multiple challenges, can be key contributors to the development of a sensitive concept of communities' resilience as they are able to indicate how their contexts address problems either by bringing in different resources, or developing strategies, by mobilising social, cultural, and intercultural capital or creating opportunities. Therefore, the study, which supports the development of the scale, provides topical elements for a better understanding of resilience indicators when considering youth-orientated community development.

Research examining resilience at the community level has developed an approach that incorporates the context and relationship between systems into the discussion [48]. These new approaches, which are more sensitive to the social, cultural, and geographical aspects of resilience, can contribute to a new understanding of the variations in resilience and the variability of factors that can be associated with resilience. This aspect is important for studying the educational pathways and lives of young people in border regions, as we argue that resilience, as a multidimensional construct, may be culture-dependent [49]. The work of Kirmayer and colleagues [50, 51] on the indigenous population of Canada, migrants and refugees advocates the concept of resilient communities by integrating people's locally rooted perceptions and reflections.

## 2. Methods

This study took place in the context of border regions, and some of aspects of the scale were based on a previous qualitative study which, among other things, highlighted the importance of culture and heritage among young people and the population at large [4].

Elsewhere we have shown that in some border regions, examples of a resilient community can be found through: (i) the development of a strong sense of belonging to the region; (ii)

regular and ritual activities, mainly related to heritage and traditions; as well as (iii) systematic support of different generations [4].

The CRS-Y scale is part of a longer questionnaire developed by the authors [52] within the project "Grow.up–Grow up in Border Regions in Portugal: young people, educational pathways and agendas" (ref. ptdc/ced-edg/29943/2017) and its contents regarding ethical issues and data privacy were approved by the Monitoring Platform for School-based Inquiries (MIME in Portuguese). The longer questionnaire is composed by four independent scales aiming to measure young people's perceptions regarding schools' resilience; communities' resilience (CRS-Y); the sense of belonging to school and the sense of belonging to the community.

## 2.1. Development of the Community Resilience Scale for Youth

To develop the scale, we conducted a study with a five-step design, as shown in Fig 1.

As suggested by the literature, to design the CRS-Y we started by identifying the theoretical basis of the construct. Designing a conceptual framework of community resilience for youth by means of a comprehensive review of literature, we confirmed that there were no similar scales available. We analysed five scales that measure resilience involving young people/adolescents in different scientific fields, particularly in health and psychology: the Adolescence Resilience Questionnaire (ARQ) [53], that measures resilience among adolescents with chronic illness; the Resiliency Scales for Children and Adolescents (RSCA) [54], which assesses the relative strength of aspects of personal resiliency as a profile in children and adolescents; the Adolescence Resilience Scale (ARS) [55], that measures psychological features of resilient individuals; and the Resilience Scale for Adolescents (READ) [56], which measures protective factors associated with fewer depressive symptoms among adolescents. The Child and Youth Resilience Measure (CYRM) [57] was a relevant instrument, as it is a culturally and contextualised measure that we also used in the development of our scale, but the focus was on youth resilience across cultures. We also explored instruments regarding community resilience such as the Community Resilience Self-Assessment [32] and the Communities Advancing Resilience Toolkit (CART) [58].

Although these latter instruments served as a source of inspiration due to their self-assessment and practice-oriented features, as well as their inclusion of dimensions of community resilience as community capitals, the first instrument focused on the resilience of forest-dependent communities, and none were youth-centred or appropriate enough to be used by young people. Our goal was to develop a scale on the dimensions of community resilience that

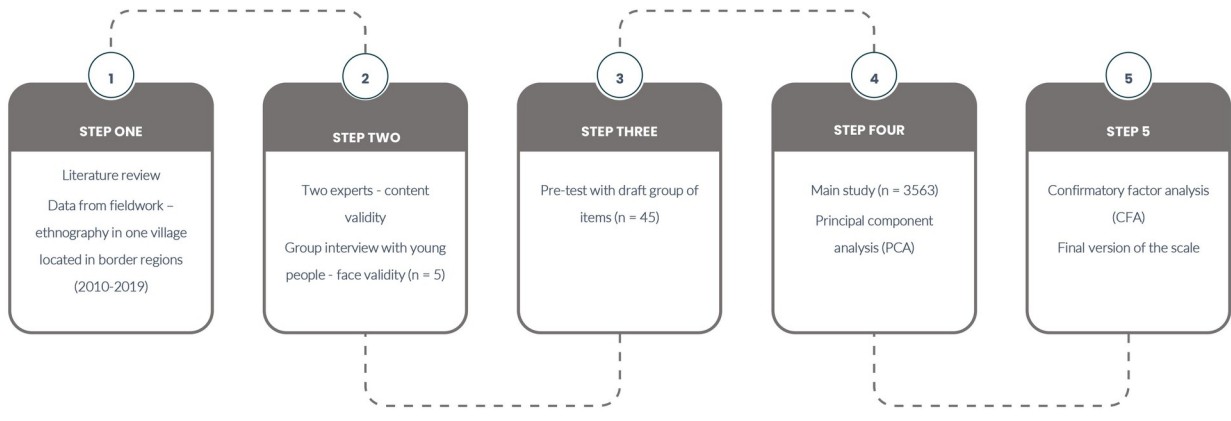

**Fig 1. Development and validation of the CRS-Y.**

focused specifically on youth development and livelihoods and that young people could respond to by sharing their perceptions of how their communities are responding to their needs. We did not find a suitable scale for this purpose.

Consequently, an initial pool of items reflecting the multidimensionality of the construct of resilient communities was generated both by a deductive and inductive approach. Deductively, through literature review, including prior research and published work by authors 1 and 3, we established a preliminary definition of the domain that included dimensions conceptually associated with resilient communities, such as reciprocity, trust, and collective action [32, 41, 47, 59–61], ties, shared values and protection [50, 62–64], social and community capital [7, 13, 39, 42, 44] and network and bridging dynamics [45, 65–67]. The inductive approach was done through interactive discussions among the project team and qualitative data gathered through an ethnographic study with young people from Portuguese border regions from 2010 to 2019 carried out by author 1 [4, 5, 68]. The ethnographic data confirmed the attributes of resilience provided by the theory and brought forward other aspects which are relevant to understanding youth trajectories in contexts of vulnerability. Thus, this data highlighted youth perspectives regarding resilience elements that are rooted in the community, as the significance of being connected to their place; the importance of caring for their roots and ties; the relevance of border crossing to gather new experiences; and the significance of engaging in intergenerational support initiatives [4, 5, 68]. Similarly to other ethnographic studies on resilience [69], the fieldwork experience helped us to develop a culturally grounded measure by revealing the voices of these young people and their communities and elucidating how structural vulnerabilities are working.

To strengthen content validity we involved two experts, one from the field of education and another from the field of psychology, who reviewed the 16 items and suggested changes, rejection, or acceptance until a consensus was achieved. The scale's content validity was reinforced by author 1's expertise on youth from vulnerable social contexts, particularly from border rural regions, as well as on resilient schools and communities [4, 5, 68, 70]. The ethnographic study in particular provided a situated understanding of resilience that benefited from in-depth knowledge about young people's feelings regarding their community and their community's resilient approaches supporting young people, aspects that are reflected in the content of the items. This previous fieldwork in a context that was part of the main study helped develop concrete and relevant items, capable of capturing young people's lived experiences and, therefore, of ensuring observable contents [71]. Additionally, to ensure face validity of the content, we asked five random students from border regions for their insights regarding the content and formatting of the items, and there were no additional suggestions or comments, with the items remaining the same.

To proceed with further development of the scale, we administered a draft of the survey items among a group of young people (n = 45), with similar characteristics to the target population of the scale, to answer and interpret items so that we could check clarity, readability, timing, and completeness. After this phase, we defined a scale with sixteen items to be used in the main study for validation.

## 2.2. Main study and scale validation

Schools selected from the border regions were contacted and the questionnaire was distributed locally, during classes, to ensure higher response rates and so that any doubts of the young people could be clarified. Data were collected during the period 2017–2018. Young people who agreed to participate signed the informed consent as well as their parents or legal guardian in case of minors.

## 2.3. Participants

This research was carried out in all the thirty-eight municipalities of mainland Portugal that border Spain (Fig 2).

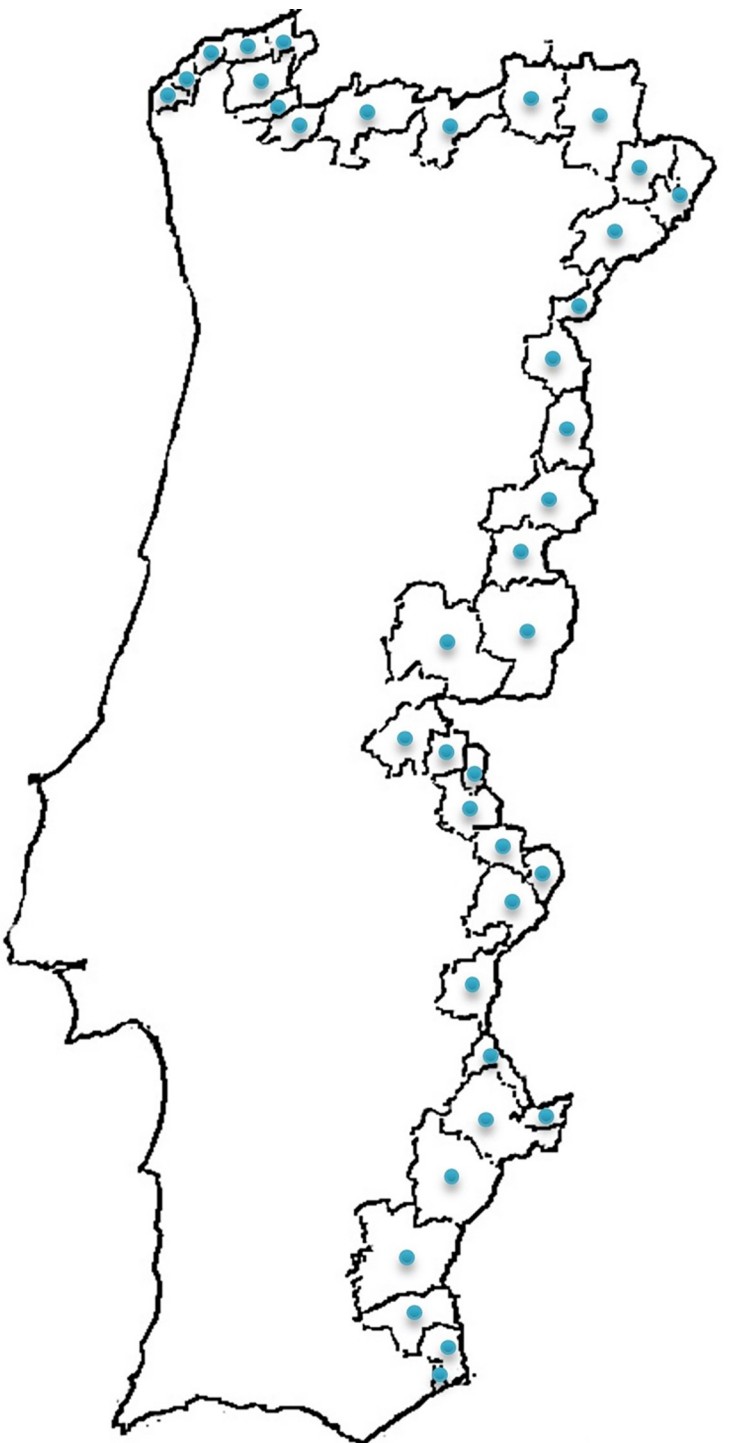

**Fig 2. Research contexts (38 municipalities).**

In thirty-four municipalities, the questionnaire was delivered in the School Cluster with secondary education that existed in each context. The structure of the Portuguese school system is mainly organised into school clusters. A school cluster offers levels of education from pre-school education to upper secondary education. Regarding the four municipalities with more than one school cluster with secondary education, one school was chosen at random. The fact that almost all secondary schools of the Portuguese border regions took part in this study certainly endows it with unusual robustness regarding the sample (and sampling error). Additionally, the high number of respondents (almost 4000; details below) grants yet another layer of confidence in the results, particularly with respect to the stability of the proposed factorial solution [72].

Once the 38 schools had been selected, the research team contacted the schools to present the study and to explore the potential interest in taking part in the project. The selection of pupils to complete the questionnaire was the responsibility of each school, bearing in mind that young people from the ninth, tenth, eleventh and twelfth grades should be able to participate in the project. Written informed consent was obtained from all the young people participating and from their parents in the case of being mature minors. This process will be fully described in the ethics statement section later in this manuscript.

This study involved 3563 young people from the Portuguese border regions from 38 school clusters (Table 1). Regarding geographical distribution by region, 56.8% of the young people were from the north of Portugal, and 31.5% from the south. The northern region has a higher demographic density, which explain a higher representativeness at this region.

These young people are from 10 schools clusters up to ninth grade (lower secondary education) and 28 school clusters up to up to twelfth grade (upper secondary education), making a total of 38 schools. In terms of demographics, the sample included 55% girls and 45% boys. Most of them (52.6%) were between 16 and 18 years old, 43.6% were between 13 and 15 years old and only 3.7% were over 18 years old.

As far as educational pathways are concerned, 32.4% of young people were in ninth grade, 28.3% in tenth grade, 21.1% in eleventh grade and 18.1% in twelfth grade. Of the young people who attended secondary school, 92.1% attended general secondary education and 7.2% attended vocational secondary education.

In terms of socio-educational indicators, 48.5% of these young people have at least 51 books at home. In terms of the mother's education, 33.3% of mothers completed secondary education, 29.6% completed basic education, 21.8% completed higher education, 5.7% completed primary education and 0.2% completed less than one year of education. Regarding the father's education, 38.7% completed basic education, 23.8% completed secondary education, 13.9% completed higher education, 9.5% completed primary education and 0.3% completed less than 1 year of education. In general, mothers have higher levels of education when compared with fathers.

This sample responded to our goal of obtaining a group of young people (9th and 12th grade) from the Portuguese municipalities located in border regions, ensuring that all border contexts were included.

## 2.4. Statistical analysis: Selected techniques and justification

The main aim of the present article is to present the development and validation of a Community Resilience Scale for Youth (CRS-Y).

Analyses were conducted with the support of R software, as well as IBM SPSS Statistics 26 and IBM SPSS AMOS 26 Graphics.

As stated above, prior research conducted by one of the authors [4, 5, 68, 69] was key in ensuring content validity, on top of a thorough literature search—based on the most

**Table 1. Participants' demographics and descriptive statistics (n = 3563).**

| | | n | % | | | n | % |
|---|---|---|---|---|---|---|---|
| **1) Region** | North | 2023 | 56.8 | | | | |
| | Centre | 418 | 11.7 | | | | |
| | Alentejo (Centre South) | 975 | 27.4 | | | | |
| | Algarve (South) | 147 | 4.1 | | | | |
| **2) Sex** | Female | 1950 | 54.7 | **3) Age** | 13–15 | 1555 | 43.6 |
| | Male | 1613 | 45.3 | | 16–18 | 1874 | 52.6 |
| | | | | | >18 | 131 | 3.7 |
| | | | | | NR | 3 | .1 |
| **4) School year** | 9th | 1153 | 32.4 | **5) Course attended** (n = 2403) | Scientific-humanistic | 2214 | 92.1 |
| | 10th | 1007 | 28.3 | | Professional | 174 | 7.2 |
| | 11th | 751 | 21.1 | | NR | 15 | .6 |
| | 12th | 645 | 18.1 | | | | |
| | NR | 7 | .2 | | | | |
| **6) Number of books** | 0 | 72 | 2.0 | | | | |
| | 1–10 | 518 | 14.5 | | | | |
| | 11–50 | 1200 | 33.7 | | | | |
| | 51–100 | 787 | 22.1 | | | | |
| | > 100 | 941 | 26.4 | | | | |
| | NR | 45 | 1.3 | | | | |
| **7) Mother's education** | No schooling | 8 | .2 | **8) Father's education** | No schooling | 12 | .3 |
| | 1–4 years | 202 | 5.7 | | 1–4 years | 338 | 9.5 |
| | 5–9 years | 1057 | 29.6 | | 5–9 years | 1379 | 38.7 |
| | 10–12 years | 1187 | 33.3 | | 10–12 years | 849 | 23.8 |
| | University | 776 | 21.8 | | University | 495 | 13.9 |
| | NR | 333 | 9.3 | | NR | 490 | 13.8 |
| | | 3563 | | | | 3563 | |

Abbreviations: *NR*, No Response.

authoritative national and international databases (e.g., EBSCOhost, Online Knowledge Library, Web of Science)—and review.

For the analysis, the overall sample of nearly 4000 subjects was randomly split into two sub-samples of (approximately) equal size (subsample A, n = 1828; subsample B, n = 1735). A principal component analysis (PCA) with varimax rotation was conducted with subsample A in order to identify the scale components. A cut-off point of .4 was used for items' communalities. Similarly, items' loadings on their respective factor were considered acceptable above .4.

After a theoretically and empirically sound factorial solution was achieved, internal consistency of the factors was examined through Cronbach's alpha and McDonald's omega. Following Nunnally's [73] authoritative source, alpha was considered acceptable above .7, but scores above .8 were desirable. Regarding McDonald's omega coefficient, values above 0.7 are considered recommendable [74].

Subsequently, confirmatory factor analysis (CFA) was conducted with both subsamples (i.e., subsample A, used to run the PCA analysis, and subsample B), in order to access the adequacy of several fit indexes, adding empirical evidence of the scale's construct validity and its replicability across samples. According to the best current scientific standards, several measures of goodness of fit (CFI, comparative fit index; TLI, Tucker-Lewis index; RMSEA, root

mean square error of approximation; and SRMR, standardised root mean square residual) are provided [74–79].

Lastly, internal consistency for the subsample B was also calculated.

## 2.5. Ethical declaration

For the study of the psychometric characteristics of this scale, we considered ethical issues essential to the development of research. Our ethical concerns began from the construction of the questionnaire: we tried to design a questionnaire with which young people could identify and in which they could feel their realities are reflected. In this regard we adjusted the language to be more suitable to young people and used language which is sensitive to gender issues. The data collection for the validation of this instrument was carried out with the ethical approval of MIME—Monitoring of Research in Education Environments–of the Portuguese Directorate-General for Education (MoE). This body of the MoE ensures that respondents are protected from any harm the questionnaire may cause, namely loss of anonymity.

Written informed consent was obtained from all young people responding to the questionnaire, and, in case of mature minors, from parents or a legally recognised surrogate decision maker. The informed consents used simple and age-appropriate information. As in context-based research children and young people are approached as a group in context-based research, we ensured that they were aware of the option not to participate. In cases in which young people did not want to participate or when we did not obtain parental approval, teachers ensured they were enrolled in other activities. This was a rare situation and the researchers and school stakeholders handled it case by case.

During the application of the questionnaire, the researchers informed young people about the goals of the study, emphasising the voluntary nature of their participation and ensuring that participants understood the informed consent. The anonymity and confidentiality of the data were guaranteed, respecting the protection of personal data as required by MIME.

This study is also characterised by a mission of social and scientific justice in approaching more remote populations and collecting data in context.

## 3. Results

### 3.1. Principal component analysis and internal consistency of random subsample A

A principal component analysis (with varimax rotation) was conducted with the initial pool of 16 items. Three components yielded eigenvalues above 1, explaining approximately 55% of the common variance. Nevertheless, some items did not contribute to a clear and empirically defensible structure, showing low communalities and/or low or cross loadings (i.e., approximate loadings in more than one of the components). Communalities below .4 were considered low [72, 75].

Loadings were considered problematic below .4, as we aimed for loadings (at least) above .5 [72, 75]. Items were considered to present cross loadings when the difference between the loadings on different components did not exceed .200 (i.e., 4% of the explained variance). Consequently, through an iterative process of removing one item and reassessing the overall factorial structure and empirical indicators, the following items were removed: item 12, ("In my region, not everyone has access to the existing cultural offer"), due to low communality (below .2), as well as low and cross loadings; item 6 ("In my region there are many job opportunities") due to low communality (below .4), as well as low and cross loadings; item 13 ("There are local initiatives to foster the history and culture of the region") due to low communality (below .4),

**Table 2. Results of principal component analysis—CRS-Y Scale (random sample a; n = 1828).**

| | | Comp. 1 | Comp. 2 | Comp. 3 | Communalities |
|---|---|---|---|---|---|
| 1 | There are several initiatives in my region to help young people in their life paths. | .704 | | | .593 |
| 2 | In my community, I have opportunities to organise useful actions (e.g., awareness campaigns, volunteering). | .779 | | | .648 |
| 3 | The adults in my community would participate in actions organised by the youth. | .759 | | | .668 |
| 4 | In my community, there are opportunities for young people to participate in decision-making processes. | .763 | | | .647 |
| 5 | There are opportunities to participate in local initiatives. | .722 | | | .624 |
| 6 | There is a lot of help among the people in my community. | | .687 | | .582 |
| 7 | In my community, people are accepted equally, regardless of ethnicity, gender or other differences. | | .781 | | .635 |
| 8 | In my region, there is a very strong investment in education. | | .651 | | .547 |
| 9 | The people of the community show concern about the departure of young people from the region. | | .661 | | .494 |
| 10 | The Portuguese and Spanish communities organise joint events. | | | .831 | .726 |
| 11 | There are people from Spain that are part of my community. | | | .799 | .656 |
| 12 | My school organises joint activities with schools in Spain. | | | .711 | .563 |
| | **% Variance** | 40.478 | 12.389 | 8.655 | -- |
| | **% Variance after rotation** | 26.106 | 18.940 | 16.476 | -- |

as well as low and cross loadings; and item 11 ("In my region there are several support infra-structures/services for young people") due to cross loadings. The low contribution of these items can be explained on the grounds of representing factual knowledge that young people might not have. Hence, the final structure is composed by three components and twelve items, as shown in the table below (Table 2). The solution accounts for 61.5% of the variance and all items present communalities above .5.

Extraction method: principal component analysis.

Rotation Method: orthogonal rotation (varimax).

Regarding internal consistency (Table 3), all components show good to excellent internal values [73, 75]: the first component, composed by five items, yields a Cronbach's alpha of .86; the second component, with four items, .73; the third component, .72 (three items). Additionally, there is no case in which the internal consistency would improve as a consequence of an item being removed.

## 3.2. Confirmatory factor analysis and internal consistency of random sample B

As an additional step to further validate the proposed structure of the CRS-Y, we conducted a confirmatory factorial analysis (with a maximum likelihood estimator) on a separate dataset (random sample B) from the one used for the principal component analysis. As show in the table below, the values of the different indexes clearly support the adequacy of the proposed structure. It is important to stress that these values were achieved without any errors' correlation or the inclusion of any additional parameters to the ones identified in the picture below (Table 4, Fig 3). Specifically, CFI and GFI present values of .98 both, significantly surpassing

**Table 3. Results of Cronbach's alpha reliability and Omega—CRS-Y Scale (random sample a; n = 1828) (random sample b; n = 1735).**

| Factor | Number of items | Random sample a | | Random sample b | |
|---|---|---|---|---|---|
| | | Cronbach's alpha | Omega | Cronbach's alpha | Omega |
| **Factor 1** | 5 | .856 | .86 | .856 | .86 |
| **Factor 2** | 4 | .734 | .74 | .744 | .75 |
| **Factor 3** | 3 | .722 | .73 | .691 | .72 |

**Table 4. Goodness-of-fit indicators of the CFA model, three factors (random sample b; n = 1735).**

| CFI | GFI | RMSEA [IC] | SRMR | χ2 | gl |
|------|------|------------------|-------|---------|-----|
| .979 | .981 | .042 (.036-.048) | .0234 | 206.407 | 51 |

the common cut-off value of .90 for good fit and the .95 threshold of an excellent fit [49, 52]. The RMSEA index also yields a good fit (.042, with the upper value of the 90% confidence interval rounding to .05), below the cutoff value of .05, as well as the SRMR. (.023), also below the .05 threshold [76, 77]. Regarding the $\chi^2$ statistic, it is important to recall the common admonition regarding its dependence on the sample size, namely given our unusually large sample (1828 subjects, and 1735 subjects, each sample) [78, 79].

The values of the standardised regression weights are significant and range from .55 to .86. In addition, the values of the correlations between the three factors were positive and statistically significant. The positive correlations we found between factors are consistent with the literature on resilient communities as building mutual trust within the community and with other communities has an impact on strengthening ties around shared values, factors related with social capital that co-shape social cohesion and, therefore, resilience.

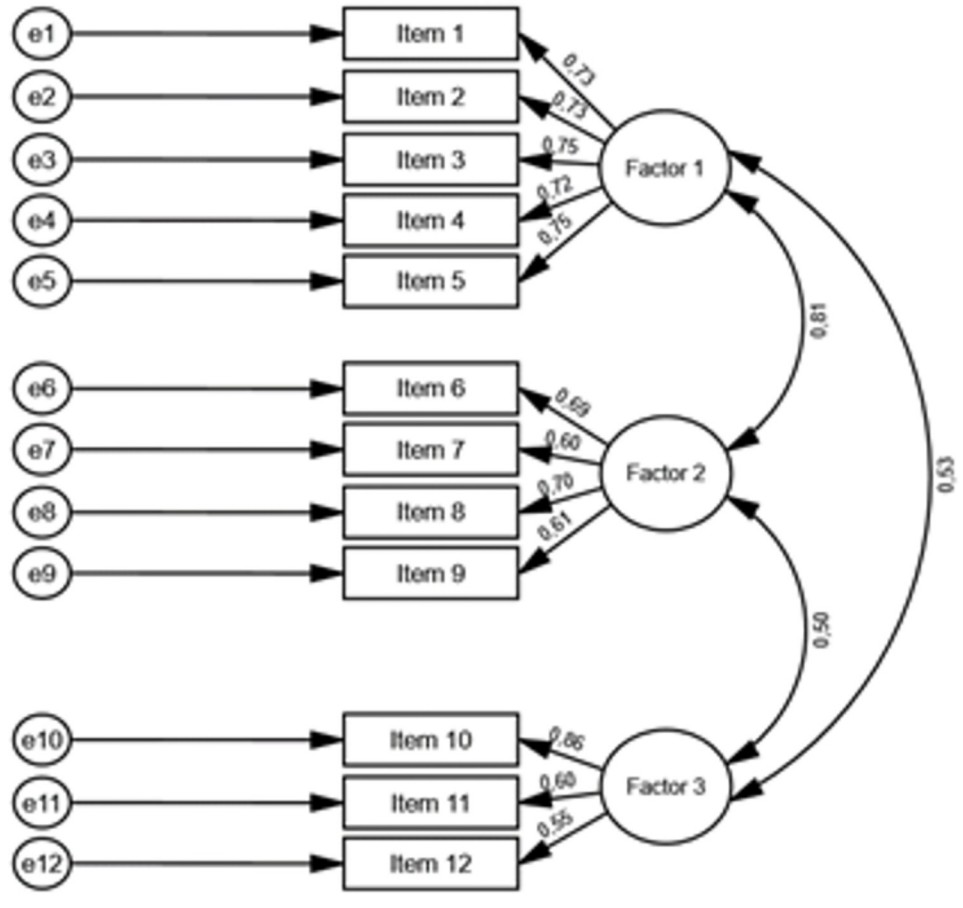

**Fig 3. Confirmatory factor analysis structure and regression/correlation values (random sample b; n = 1735).**

In short, the indexes produced by the confirmatory factor analysis confirm the appropriateness and adjustment of the proposed three-factor solution for the CRS-Y.

As the correlation between factors 1 and 2 was high (.81, please see Fig 3), we performed a confirmatory factor analysis (maximum likelihood estimator) with two factors, merging factor 1 and factor 2 together. The results, as seen in Table 5, are considerably worse than the three-factor model, namely regarding the RMSEA values, significantly surpassing .05. Therefore, we conclude that the three-factor solution is, indeed, theoretically and empirically sound.

**3.2.1. Factor 1: Promotion of opportunities and collective trust.** Resilient communities are those that expand or create spaces in which individuals can act and be recognised as co-designers and resources and not just as victims of problems. When focusing on young people, the literature considers that recognition and appreciation of young people's strengths and potential can promote young people's collective self-esteem and empowerment, a fundamental construct of youth development [80]. It also confirms that participation is a relevant factor for community cohesion and trust, as well as positive youth development. Participation in the community is a "vital synergy among community capital" [39] (p. 939), which makes it critical to assess how a community is able to involve young people in community life [62].

When young people understand that they have the support of their community, especially of adults, this has a positive effect on their well-being and recognition and can act as a protective factor. Unequal distribution of power can be an obstacle to young people participating fully in their community [81]. A community that values and participates in the activities promoted by young people shows a long-term accumulation of trustworthiness, reciprocity and recognition. When young people perceive that the community supports their life' pathways, it means that they evaluate the existence of structural behaviours, a sustainable and collective commitment to a common goal, rather than a rare event.

**3.2.2. Factor 2: Promotion of shared values and protection.** Many elements can act as promising roots of resilience, namely community-dynamic interactive processes [82] and inclusive practices that can contribute to encouraging constructive behaviours and support young people's ability to critically overcome obstacles. The feeling that the community is concerned with young people's future works as a protective factor for positive youth development, particularly when young people perceive that there is a strong investment in education.

The promotion of collective protection and cohesion against threats, such as fewer opportunities and depopulation, are resilience indicators. It refers to the capacity of a community to create opportunities for building the capacity of individuals [39] as providers of education opportunities.

The literature points to the ability to anticipate and develop social and cultural practices as bonding strategies to keep young people attached to the region as relevant characteristics of resilience [14, 59]. A resilient community has shared commitments and values, sometimes based on emotional dimensions or interests [63]. Aspects such as an appreciation of cultural identity can foster a sense of belonging [64], and a collective way of thinking [83] can strengthen the resilience of a community.

For the understanding of youth development, especially regarding participation, educational pathways and transitions, it is fundamental to develop a concept of resilience as a "community and cultural process" [46] (p. 62) that depends more on networks and collective experiences, systemic and organic changes than on a single factor and individual processes.

**Table 5. Goodness-of-fit indicators of CFA model, two factors (random sample b; n = 1735).**

| CFI | GFI | RMSEA [IC] | SRMR | $\chi2$ | gl |
|---|---|---|---|---|---|
| .945 | .953 | .066 (.061-.072) | .0375 | 456.211 | 53 |

### 3.2.3. Factor 3: Promotion of intercommunity trust and ties.

Cross-border social relationships foster intercommunity trust and reciprocity, which is a type of social capital [60], and are fundamental in solving shared salient problems among different communities such as an environmental disaster, economic crisis or pandemic. In fact, some border regions of Spain and Portugal have informal networks and many border regions are spaces of interdependency. Informal networks are considered relevant for social integration and cohesion in borderlands [65].

Social networks are a bridging key factor in fostering social cohesion and trustworthiness and, therefore, a resilient community. Bridging social capital grounds in ties developed among different individuals and communities and contributes to expanding circles of trust [67].

Trust is a learned capacity grounded in everyday life experiences and socialisation. The existence of shared organisation of events and regular contact with people from the neighbouring country that are not part of the immediate circle are examples of collaboration and trust-building experiences, that may foster community resilience. Being exposed to diversity and developing positive cross-border relationships are key factors for the promotion of bridging social capital [32], of generalised trust and, in consequence, increasing community social capital. This creates a particular environment where young people are growing up and becoming citizens of a larger community than their own, benefiting from cross-border cohesion.

Many young people growing up in those regions, being born after the Schengen Area was established when border controls were eliminated, often develop social interactions with population from Spain and they often have friends and relatives living across the border [5]. Their perceptions about the existence of cross-border ties and interconnections brings evidence about contact with diversity networks with different communities.

## 4. Discussion

This study looks at young people's perceptions and what they have identified as stronger or weaker in their community in terms of opportunities that meet their specific needs. The study refers to observable characteristics that develop a resilient community into a strong collective body that also connects with differing realities and communities. Cross-border informal networks may increase trust, solidarity, shared opportunities, and values may have a relevant impact on young people's development.

In this study, the factor analysis identified three factors for measuring the resilience of communities, in terms of youth development, contributing to strengthening the CRS-Y's content validity. These include young people's perceptions about the existence of opportunities, recognition and trust, as indicated by factor 1, or perceptions about being protected and belonging to a community committed to shared values and priorities, as shown by factor 2, and the perception of being connected with outside communities, through cross-border ties. The literature confirms these as dimensions adequate to assess the resilience of communities. Items confirm the idea of the connectedness of individuals, and the shift from an individual approach to one that also considers cultural, social and economic components to be important [46].

The scale makes it possible to reflect on aspects of resilient communities which are relevant for youth development: community competence [13], visible in the ability to anticipate and consciously shape a collective focus and collective action; social capital [13, 64, 84], visible in social support, networking, community participation and mutual trust [42, 83]; and engagement and cultural ties [64], as aspects related to collective self-esteem and cultural identities [13].

Factors are based on the concept of community, which combines two aspects: a community understood as a context with risk and protective factors that influence the well-being of the individual, and a community understood as a collective actor that can demonstrate resilience within itself by responding to structural adversity [61] and that has the capacity to network with other communities. This does not mean that a resilient community has no weaknesses, but it understands them.

The three factors point to efforts to promote young people's commitment not only to the region in a larger sense, but also to education and the future, even if this means that young people might leave their home region. A community's attitudes towards the future of young people can be an indication that efforts and networking are addressing a common goal or priority: "Such networking allows communities to form a common cause and to find resources and share experiences in ways that may confer new types of resilience" [46] (p. 65).

The scale includes indicators of social capital as reciprocity, trust and co-operation [40, 42] and indicators of a community's capacity to become resilient or develop resilient approaches [66, 85]. This is particularly relevant for the types of contexts that have inspired, but not limited to, the development of this scale. A community can have different meanings, sizes and levels. In less densely populated areas with scarce resources and opportunities, the role of a community and leaders is fundamental to shaping a social path. The survival of these communities depends on their networks, including cross-border networks, and how individuals build trust and a sense of belonging.

The contribution of Adger [7] to the inclusion of the value of social capital confirms the three factors:

> social capital describes relations of trust, reciprocity, and exchange; the evolution of common rules; and the role of networks. It gives a role to civil society and collective action for both instrumental and democratic reasons and seeks to explain differential spatial patterns of societal interaction. (p. 389)

The scale covers aspects related to the capacity of a community to meet the expectations of young people. The emphasis on participation opportunities is due to the lack of recognised spaces for young people to express their aspirations. Therefore, this aspect of social capital, which promotes engagement, together with reciprocity and trust, is fundamental for young people's development.

## 5. Conclusions

The development of a Community Resilience Scale for Youth aims to ensure fairness in assessing, through young people's perceptions, the capacity of a community or region to address the problems, interests and priorities regarding young people growing up in peripheral contexts. For this reason, it is a youth-oriented community resilience scale. The more equipped a context is for developing components related to resilient communities, the more we expect young people to be more engaged and purposeful. Better information about which resilience components are missing in a given place will help in designing and implementing youth policies and practices.

This scale, which focuses on youth development at the regional level, introduces elements for discussing resilience thinking [59] within governance planning systems, which are usually more influenced by short-term responses to immediate concerns rather than long-term approaches. Some of the problems related to youth, especially vulnerable contexts, are not solved by one-off solutions or in isolation. Understanding young people's perceptions on what

is available in their communities to support their experiences and life choices navigation is important for stakeholders to balance a diversity of capitals that work best for a particular community and in a particular situation. Short-term commitments and obligations limited to a restricted area will have limited impact.

We recognise that youth development is only partially influenced by the resilience of organisations, communities, or regions, and by practices and resources which are dependent only on the community. Nevertheless, we consider that community-focused policies will help to promote resilience in many sectors of the community:

> Community resilience, therefore, is often associated with the quest for multiple resiliencies within a community pursued by highly varying stakeholder networks, some of which may be directly contradicting and undermining efforts by other groups in the community to achieve maximum resilience. [85] (p. 1219)

This scale proposal is based on the idea that resilience is not a characteristic that an individual or a community has or does not have, but as the result of a complexity of factors and combinations of factors.

Considering that the construct of resilient communities needs to involve long-term and solid approaches, this scale can help to indicate the maturity of regional or municipal policies to meet young people's needs and set priorities. The resilience of communities, in a process-orientated approach to changing and challenging problems, works not only through adaptation/learning models but also through a strategy to respond, disrupt and influence systemic change [32].

We believe that working on resilience does not only depend on the researchers' ability to evaluate all principles. Associated with this is a precise (also epistemological) process of investigating the social conditions underlying the possibilities for a response. This is important for access and planning. Drawing on the vulnerability paradigm [29], we try to understand "how social, economic, and political relations influence, create, worsen, or can potentially reduce hazards in a given geographic location" (p. 17).

The scale refers to the resilience of communities in promoting youth development and significant pathways. It allows to understand what the ideal conditions are for promoting young people's well-being, quality of life and purposeful pathways. When used longitudinally, the scale can be helpful in understanding how a resilient community is confronted with conditions that change over time, with a critical event with profound effects, and with structural and more permanent challenges, such as unemployment or isolation. In this case, we would have access to trajectories of resilience.

We believe that this scale can meet the needs of a variety of communities and contexts, as well as practitioners and policymakers responsible for the development and well-being of young people.

## 6. Limitations and future research

A limitation of the scale could be the fact that the study is based on a sample of Portuguese young people. The replication of the study in other countries would provide an additional understanding of its intercultural relevance. Future studies with other groups of young people from Portugal will allow exploration of the three-factor structure.

## Author Contributions

**Conceptualization:** Sofia Marques da Silva, Gil Nata.

**Data curation:** Ana Milheiro Silva, Sara Faria.

**Formal analysis:** Gil Nata, Sara Faria.

**Funding acquisition:** Sofia Marques da Silva, Ana Milheiro Silva.

**Investigation:** Sofia Marques da Silva, Ana Milheiro Silva.

**Methodology:** Sofia Marques da Silva, Gil Nata, Ana Milheiro Silva, Sara Faria.

**Supervision:** Sofia Marques da Silva, Gil Nata.

**Writing – original draft:** Sofia Marques da Silva.

**Writing – review & editing:** Sofia Marques da Silva, Gil Nata, Ana Milheiro Silva, Sara Faria.

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
