## [Decision Letter · Decision Letter 0]

4 Nov 2021

PONE-D-21-25006Development and validation of a Community Resilience Scale for Youth (CRS-Y)PLOS ONE

Dear Dr. Silva,

Thank you for submitting your manuscript to PLOS ONE. After careful consideration, we feel that it has merit but does not fully meet PLOS ONE’s publication criteria as it currently stands. Therefore, we invite you to submit a revised version of the manuscript that addresses the points raised during the review process.

Both reviewers address important topics to enhance the quality of the study. While one emphasized conceptual aspects of resilience as a construct, the other addressed methodological considerations that should be clarified. The revision of the academic English language must be conducted.  

We look forward to receiving your revised manuscript.

Kind regards,

Roxanna Morote Rios, Ph.D

Academic Editor

PLOS ONE

Reviewers' comments:

Reviewer's Responses to Questions

**Comments to the Author**

1. Is the manuscript technically sound, and do the data support the conclusions?

Reviewer #1: Partly

Reviewer #2: Partly

2. Has the statistical analysis been performed appropriately and rigorously? 

Reviewer #1: Yes

Reviewer #2: Yes

3. Have the authors made all data underlying the findings in their manuscript fully available?

Reviewer #1: Yes

Reviewer #2: Yes

4. Is the manuscript presented in an intelligible fashion and written in standard English?

Reviewer #1: Yes

Reviewer #2: No

5. Review Comments to the Author

Reviewer #1: The statistical analyses are thorough and well described and has potential to add to the literature on community resiliency.

But I believe this article requires clarification on the survey items before it can be published. Specifically on lines 188 forward it says the 16 items used in this study were part of a “wider questionnaire.” Were the 16 items part of an actual questionnaire from Silva SM, Silva AM (2016)? If so, why were these particular 16 items selected? A little more information about the wider questionnaire would be useful, too (and better to say “longer” questionnaire?).

Also, on line 197 forward “Additionally, the scale’s content validity [45] was strengthened by one of the authors expertise of the context, which conducted an ethnographic study with young people from Portuguese 199 border regions [4, 5].” I recognize the expertise of the authors, but more details of the content validity alluded to should be explained. The factor analysis reported in the article will further contribute to the CRS-Y’s content validity.

I would have described the 4 items deleted from the pool of 16, perhaps including them.

The description of the statistical treatment and discussion of the three factors was solid.

The description of the participants demographics was solid and informative.

I had several minor comments about specific sentences that will need clarification. These include (line 32) “bounce challenges”. Better to say “bounce off challenges?

I assume “borderlands” refers specifically to the border with Spain, and perhaps that should be made explicit sooner.

Line 109 ” However, the latter do not understand resilience ..” are “the latter” Callaghan and Colton?

Line 164 forward ” Portuguese border regions, as they face multiple challenges, can be key contributors to the development of a sensitive measure ..” is awkward as it sounds like the border regions are developing a measure.

I suspect references 43 to 45 (Cohen to Urbani) is not necessary as most readers will be familiar with basic concepts of reliability and validity.

Reviewer #2: While the paper seems to be based on a rigorous data collection, and discusses an interesting issue, that link to resilience is not clear. First, resilience is usually applied in the context of a shock. It is not clear that economic decline or lack of opportunity necessitates a resilience approach. Perhaps you wish to argue that resilience needs to be considered for slow onset or social factors such as economic decline. But then you should clearly explain why. Right now it seems to me that you could just ask youth what factors they feel are important to their inclusion or future in the community? Why even bother calling this resilience? What value does the resilience approach add?

Second, have the results been verified on the ground? Are the communities that have the three factors that you have identified more resilient? Do they have more youth, or are growing faster or bounce back from shocks better? If you have not done this, then it appears that the students answers could just be opinion about resilience but not actually an indicator of what is needed for resilience.

Third the paper should be reviewed by an English speaker for grammar and wording. In some places words are missing or the ideas are not very clear. For example, even in the abstract a word is missing.

Finally, it might be nice to have some figures, such as a map of the areas sampled or figures of the statistical results.

6. PLOS authors have the option to publish the peer review history of their article (what does this mean?). If published, this will include your full peer review and any attached files.

Reviewer #1: No

Reviewer #2: No

---

## [Author Response · Author response to Decision Letter 0]

16 Jan 2022

Dear Reviewers

Thank you for the valuable time spent in reading our manuscript and for the useful inputs that made us reflect further and contributed to improve the manuscript.

For clarity we copied each comment from each reviewer and we add our reply immediately after each point.

The revised manuscript has highlights in yellow indicating all the additions and changes we have made to the manuscript.

Responding to Reviewer 1

Comment 1: “But I believe this article requires clarification on the survey items before it can be published. Specifically on lines 188 forward it says the 16 items used in this study were part of a “wider questionnaire.” Were the 16 items part of an actual questionnaire from Silva SM, Silva AM (2016)? If so, why were these particular 16 items selected? A little more information about the wider questionnaire would be useful, too (and better to say “longer” questionnaire?).

Authors' response: detailed information regarding the procedures of the scale development and validation, as item generation, content (by experts) and face validity (by young people) was included in the section 2.1. “Development of the Community Resilience Scale for Youth”. (page 10, 11 and 12). 

For a clear understanding of the process, we included the figure a new figure (Fig.1) with the five-step design we carried out. The figure is to be inserted in page 10.

In the introduction to the Methods section (2. Methods, page 9), we included some more information about the wider questionnaire.

Comment 2: “Also, on line 197 forward “Additionally, the scale’s content validity [45] was strengthened by one of the authors expertise of the context, which conducted an ethnographic study with young people from Portuguese 199 border regions [4, 5].” I recognize the expertise of the authors, but more details of the content validity alluded to should be explained. The factor analysis reported in the article will further contribute to the CRS-Y’s content validity.”

Authors' response: regarding the content validity, we included more information about the process of the validity of the items in the non-statistical part of the study, that is, in items formulation. We explained the rational under the generation of the items that cover dimensions that indicate communities’ resilience. Therefore, we tried to be clearer in the process of content validity, by indicating the participation of two experts, in particular to assess if the proposed items were adequate to measure communities’ resilience for youth development. We tried to be more convincing about the value of the expertise of author 1 in studying youth in vulnerable contexts, youth in border regions and resilience and how it contributed to reassure content validity.

Thank you for this comment and apologies for the lack of clarification on this matter in the manuscript.

Comment 3: I would have described the 4 items deleted from the pool of 16, perhaps including them.

Authors' response: the four items deleted from the pool of sixteen were included. A justification and explanation regarding each item elimination were also added, such as low communality and low and cross loadings. This information is included in page 17 around line 399.

Comments 4: These include (line 32) “bounce challenges”. Better to say “bounce off challenges?

Authors' response: the sentence was corrected

Comment 5: I assume “borderlands” refers specifically to the border with Spain, and perhaps that should be made explicit sooner.

Authors' response:we have included the clarification in the abstract (page 1) and in the introduction (page 2), highlighted in yellow.

Comment 6: Line 109 ” However, the latter do not understand resilience ..” are “the latter” Callaghan and Colton?

Authors' response: we have clarified the author to whom we are referring to, by rewriting the sentence. (page 6, line 132)

Comment 7: Line 164 forward ” Portuguese border regions, as they face multiple challenges, can be key contributors to the development of a sensitive measure ..” is awkward as it sounds like the border regions are developing a measure.

Authors' response:we changed the sentence by removing the word measure and adding additional clarifications. What we wanted to say is that while the authors of this manuscript are those who developed the measure, social actors from the context may contribute through their situated knowledge to the development of items/indicators. Thank you. It was an odd sentence.

Comment 8: I suspect references 43 to 45 (Cohen to Urbani) is not necessary as most readers will be familiar with basic concepts of reliability and validity.

Authors' response: the references were deleted.

Responding to Reviewer 2

Comment 1: While the paper seems to be based on a rigorous data collection, and discusses an interesting issue, that link to resilience is not clear. First, resilience is usually applied in the context of a shock. It is not clear that economic decline or lack of opportunity necessitates a resilience approach. Perhaps you wish to argue that resilience needs to be considered for slow onset or social factors such as economic decline. But then you should clearly explain why. Right now it seems to me that you could just ask youth what factors they feel are important to their inclusion or future in the community? Why even bother calling this resilience? What value does the resilience approach add?

Authors' response: thank you for this comment. We included a cear justification on why knowledge on resilience and community’s resilience may still serve many purposes, namely for communities’ self-assessment and to better develop strategies do work on their strengths.

Resilience has been associated to disaster response and adaptation. This perspective, mainly from environment studies, is not the one we are using in this study. We are aligned with authors that have developed an ecological and transformative perspective of resilience. We tried to make our standpoint regarding the usefulness of the concept clearer in the Introduction of the manuscript (pages 3, 4 and 5, from around line 62 to line 102). 

 Lately, in the context of the disruption caused by the pandemic, the concept is being used politically and appropriated by the civil society which made us made us think about the social and political pertinence and uses of the concept.

Comment 2: Second, have the results been verified on the ground? Are the communities that have the three factors that you have identified more resilient? Do they have more youth, or are growing faster or bounce back from shocks better? If you have not done this, then it appears that the students’ answers could just be opinion about resilience but not actually an indicator of what is needed for resilience.

Authors' response: thank you for this comment. The items were generated with empirical and theoretical basis and reflect indicators of resilience. Young people answered based on their perceptions considering what we (authors) conceptualized as resilience. We asked them to provide us their perceptions about how their communities are “behaving” regarding indicators of resilient communities.

The verification on site of the results will be developed in the future as other types of evaluation of communities’ resilience will be considered. However, this was not the aim of this study. We included a clarification about this in page 3 and 4 explaining that our aim was to assess the existence of attributes of resilience through perceptions of young people in each respective community. 

Comment 3: Third the paper should be reviewed by an English speaker for grammar and wording. In some places words are missing or the ideas are not very clear. For example, even in the abstract a word is missing.

Authors' response: the manuscript was reviewed by a native speaker, and we also included a proofreading declaration

Comment 4: Finally, it might be nice to have some figures, such as a map of the areas sampled or figures of the statistical results.

Authors' response: in the first version we included a figure (figure 3) with the statistical results, but we were probably not clear about that. PLOS ONE guidelines indicate that figures need to be in separated files. Besides this figure, we included in this version a new figure with the map of Portugal indicating the research contexts (figure 2)

---

## [Editor Report · Decision Letter 1]

13 May 2022

Development and validation of a Community Resilience Scale for Youth (CRS-Y)

PONE-D-21-25006R1

Dear Dr. Silva,

We’re pleased to inform you that your manuscript has been judged scientifically suitable for publication and will be formally accepted for publication once it meets all outstanding technical requirements.

Kind regards,

Imelda K. Moise, Ph.D., MPH

Academic Editor

PLOS ONE
---

## [Editor Report · Acceptance letter]

12 Jul 2022

PONE-D-21-25006R1 

Development and validation of a Community Resilience Scale for Youth (CRS-Y) 

Dear Dr. Silva:

I'm pleased to inform you that your manuscript has been deemed suitable for publication in PLOS ONE. Congratulations! Your manuscript is now with our production department. 

Kind regards, 

on behalf of

Dr. Imelda K. Moise 

Academic Editor

PLOS ONE